# Unveiling the Evolution of Virtual Reality in Medicine: A Bibliometric Analysis of Research Hotspots and Trends over the Past 12 Years

**DOI:** 10.3390/healthcare12131266

**Published:** 2024-06-26

**Authors:** Guangxi Zuo, Ruoyu Wang, Cheng Wan, Zhe Zhang, Shaochong Zhang, Weihua Yang

**Affiliations:** 1Shanghai National Clinical Research Center for Endocrine and Metabolic Diseases, Key Laboratory for Endocrine and Metabolic Diseases of Chinese Health Ministry, Ruijin Hospital, Shanghai Jiaotong University School of Medicine, Shanghai 200025, China; guangxizuo0417@163.com; 2Department of Global Public Health, Karolinska Institutet, 17177 Stockholm, Sweden; sophiism2022@163.com; 3College of Electronic Information Engineering, Nanjing University of Aeronautics and Astronautics, Nanjing 210016, China; wanch@nuaa.edu.cn; 4Shenzhen Eye Institute, Shenzhen Eye Hospital, Jinan University, Shenzhen 518040, China; whypotato@126.com

**Keywords:** virtual reality, medicine, bibliometric analysis, research hotspots, trends

## Abstract

Background: Virtual reality (VR), widely used in the medical field, may affect future medical training and treatment. Therefore, this study examined VR’s potential uses and research directions in medicine. Methods: Citation data were downloaded from the Web of Science Core Collection database (WoSCC) to evaluate VR in medicine in articles published between 1 January 2012 and 31 December 2023. These data were analyzed using CiteSpace 6.2. R2 software. Present limitations and future opportunities were summarized based on the data. Results: A total of 2143 related publications from 86 countries and regions were analyzed. The country with the highest number of publications is the USA, with 461 articles. The University of London has the most publications among institutions, with 43 articles. The burst keywords represent the research frontier from 2020 to 2023, such as “task analysis”, “deep learning”, and “machine learning”. Conclusion: The number of publications on VR applications in the medical field has been steadily increasing year by year. The USA is the leading country in this area, while the University of London stands out as the most published, and most influential institution. Currently, there is a strong focus on integrating VR and AI to address complex issues such as medical education and training, rehabilitation, and surgical navigation. Looking ahead, the future trend involves integrating VR, augmented reality (AR), and mixed reality (MR) with the Internet of Things (IoT), wireless sensor networks (WSNs), big data analysis (BDA), and cloud computing (CC) technologies to develop intelligent healthcare systems within hospitals or medical centers.

## 1. Introduction

Virtual reality (VR) is a three-dimensional (3D) artificial environment representing realistic or non-realistic situations and comprises computers, human–computer interfaces, graphics, and sensor technology. It enables participants and observers to immerse themselves and interact in real time [1]. And compared with training by a person, VR has higher flexibility in time and space. This secure and real-time interactive virtual environment aids in performing many medical activities. For example, in fields such as anatomy teaching [2], surgical training [3], the treatment of children with autism [4], post-stroke rehabilitation [5], the treatment of post-traumatic stress disorder (PTSD) [6], and telemedicine systems [7], VR application produces outstanding results. With the development of VR, subcategories, such as mixed reality (MR) [8], an overlay of synthetic content that is anchored to and interacts with objects in the real world-in real time, augmented reality (AR) [9], an overlay of computer-generated content on the real world that can superficially interact with the environment in real time, and extended reality (XR) [10], referring to all real and virtual environments generated by computer technology and wearables, have been applied to promote the development of medicine. Furthermore, the emergence of a 3D digital space that blends the real and virtual worlds, the metaverse, could also significantly affect future clinical practice and human health [11,12]. The future trend is to integrate VR technology with innovative technologies such as digital twins (DTs), the Internet of Medical Things (IoMT), blockchain-based metaverse platforms, and other novel technologies to create intelligent medical care systems [13,14]. There is still a significant gap between current work and future trends, which presents new challenges and provides new directions for the development and application of VR technology in the medical field.

This study aimed to understand the application of VR technology in medicine in depth by using bibliometric methods and analyzing Scientific Citation Index (SCI) papers on VR in medicine. The data included countries or regions, institutions, study categories, keywords, and hot references. At this study’s core, a visual and unbiased method was established to explore hot knowledge frontiers in the research field. Using the research method proposed in this study, information about the distribution and influence of this research in countries or regions, institutions, and other aspects was obtained, and the research hotspots, future developments, and potential challenges of VR in the field of medicine were discussed. This will provide a reference for computer researchers, educators, and experts in the field of medical engineering.

## 2. Materials and Methods

On 23 January 2024, all citation data published from 1 January 2012 to 31 December 2023 were obtained from the WoSCC. These data were independently verified by Guangxi Zuo and Ruoyu Wang. The retrieval formula was TS = (VR or “virtual reality” or “virtual simulation” or “augmented reality” or “mixed reality” or XR or extended reality) AND TS = (medicine or medical). Only articles written in English were included in the research, and proceeding papers, review articles, book chapters, early access, editorial materials, letters, meeting abstracts, corrections, data papers, clinical trials, unspecified papers, case reports, biographies, and retracted publications were excluded. After reading the title and abstract of each paper, articles containing content unrelated to medical applications of VR were eliminated manually in order to obtain more accurate results. Exclusion criteria were as follows: (1) the research topic was not VR, (2) the research direction was unrelated to medicine, and (3) the research was unrelated to VR applications in medicine. The following basic data were collected for each publication: title, year of publication, country or region, institution, subject category, and keywords. The processing details of the retrieval and analysis are shown in Figure 1. The data included in this study are related to VR in medicine. General data, countries or regions, institutions, subject categories, references, and keywords were analyzed using CiteSpace 6.2. R2. This article includes the features of all citations.

## 3. Results

### 3.1. Distribution of Articles Using Publication Year

In total, 2143 studies focusing on VR in medicine, published between 1 January 2012 and 31 December 2023, were analyzed. The data collected and counted from WoSCC were deduplicated using the duplication removal function of the CiteSpace software.

Between 2012 and 2018, the number of annual articles on VR applications in medicine grew steadily. This number began to grow rapidly in 2019 and has surpassed 300 in 2021. Figure 2 shows the annual number of publications.

### 3.2. Countries or Regions 

This paper made use of the preset values in the CiteSpace system to generate statistics on the articles published in each country as well as the analysis of the cooperative relationships between countries and regions. The citations covered 86 countries and regions. Each green node area, as shown in Figure 3, represents the number of studies. Each individual circle is designated to represent the aggregate volume of publications for a particular year, with the magnitude of its radius serving as a direct indicator of the total number of publications. Specifically, a larger radius signifies a higher volume of publications, whereas a smaller radius corresponds to a lower volume. Furthermore, the proximity of a circle to the central point is inversely proportional to the chronological order of the year, with circles nearer to the center representing earlier years. Correspondingly, the color intensity of the circle darkens as it approaches the center, providing a visual cue for the earlier years. The top three countries with the largest green nodes were the United States, China, and England, with 461, 329, and 190 articles, respectively. Therefore, the links between nodes express the cooperative relationships between countries. Furthermore, more connected countries have more clouts. The academic leverage of a country is demonstrated by the size of the pink circle, as shown in Figure 3. The more connections there are to the node, the higher the country’s centrality and the wider the pink circle. According to the value calculated by the Citespace software, the node with the widest pink circle (0.41, i.e., centrality) is corresponded to the “USA” label, indicating that the articles published in the United States had the greatest overall influence on the application of artificial intelligence in medicine, and the data in Table 1 confirms this conclusion. The H-index is an indicator of research output and contribution and shows a strong correlation with medical academic promotion. The United States had the highest H-index for academic achievement, followed by England and Italy. As shown in Table 1, the United States generally has the maximal sum of publications and is considered the most predominant country.

### 3.3. Institutions

Table 2 lists the 10 institutions that have carried out the most research on VR medicine in the selected papers. The data displayed are processed from the default settings of CiteSpace. The University of London in England had the highest participation in relevant studies, with 43 articles included in the statistics. England has three institutions on the list, while the United States and Denmark each have two institutions on the list. In addition, institutions from Canada, France, and China were included. The nodes’ size positively correlated with the number of articles published by an institution (Figure 4). The link between the nodes reflects the collaborative relationship between institutions. The University of Toronto in Canada had the highest H-index for academic achievement, followed by institutions in the United States, England, and Denmark.

### 3.4. Research Categories

Figure 5 shows the category-based clustering generated by the CiteSpace software. The overlap of the clustering color blocks represents the connection between the subject areas.

Cluster 0: Engineering, Electrical and Electronic

This large cluster covers 25 categories that are related to engineering and the electrical and electronic fields, such as computer science, information systems, telecommunications, and software engineering, which are influential areas in virtual medicine. This cluster overlaps with Cluster 4.

Cluster 1: Surgery

This cluster, containing 24 items, including clinical neurology, radiology, nuclear medicine and medical imaging, and orthopedics, indicates the application of VR technology in medical surgery. The cluster mainly overlaps with Cluster 3 and Cluster 5.

Cluster 2: Mathematical and Computational Biology

This cluster, with 21 items, focuses on biotechnology, which is the use of an organism, or a component of an organism or other biological system, to make a product or process. This cluster has a considerable overlap with Cluster 3.

Cluster 3: Neurosciences

This cluster consists of 19 items in the disciplines of psychology, geriatrics and gerontology, psychology, and experimental physiology and overlaps with Cluster 1 and Cluster 2.

Cluster 4: Materials Science, Multidisciplinary

This cluster contains 15 items, involving physics, chemistry, engineering, and surgery, and mainly overlaps with Cluster 0.

Cluster 5: Healthcare Sciences and Services

This cluster of 13 items addresses the need to combine disciplines such as medical informatics, public environmental and occupational health, educational scientific disciplines, and health policy and services using a systematic approach. This cluster mainly overlaps with Cluster 1.

### 3.5. Keywords

To better understand VR applications in the medical field over the past 12 years, this study analyzed the emerging keywords developed over time based on the keyword co-occurrence cooperation network analysis diagram. This reflects the transfer of the research focus, which is displayed in Figure 6. The bursting keywords for the time frame under investigation are represented by the red squares in Figure 6. The emerging words during 2012–2023 include virtual reality simulator (2012–2017), acquisition (2012–2019), virtual reality simulation (2012–2017), operating room (2012–2017), trial (2012–2017), construct validity (2012–2017), residents (2013–2019), surgical skills (2012–2019), neurosurgery (2012–2015), surgical simulation (2012–2015), competence (2012–2017), learning curve (2014–2019), randomized controlled trial (2016–2019), skills (2016–2019), information (2018–2021), exposure therapy (2018–2021), task analysis (2020–2023), deep learning (2020–2023), machine learning (2020–2023), and three-dimensional displays (2020–2021).

### 3.6. High-Impact Papers

Table 3 elaborates on the top 10 citations in the articles on VR in medicine. In Table 3, articles are ranked from highest to lowest according to the number of citations.

## 4. Discussion

### 4.1. Principal Results

This study’s results indicate that the number of articles on VR applications in medicine has increased dramatically in recent years. One of the critical factors contributing to this is the COVID-19 pandemic [9], which began in the spring of 2020 and hampered many conventional diagnoses, treatments, and teaching; therefore, people completed teaching tasks and realized telemedicine services by building simulation models and virtual environments. The United States had the greatest centrality and H-index in terms of article count. This shows that the United States is a global leader in this area of study. England and Italy also demonstrated great centrality and influence. China has published a considerable number of papers; however, it lacks highly cited articles, indicating that the country is actively exploring the field of virtual medicine. The best three research institutions out of the many were England, the United States, and Canada in terms of the quantity of publications. Regarding the H-index, the University of Toronto in Canada and the University of California System in the United States have significantly impacted this field. The broad range of topics covered by the selected data in Figure 5 suggests there is positive room for expansion. Furthermore, through the burstiness and time zone analysis of keywords, we obtained and analyzed the changing direction of active topics over time. The focus of VR medicine research has shifted from the relatively single function of improving the proficiency of surgical operations and assisting the rehabilitation of stroke patients to a combination with artificial intelligence to solve complex clinical tasks, such as the construction of skull defect prostheses [25] and the delineation of radiotherapy target areas [26]. VR provides a strong sense of immersion and presence, which can improve telesurgery effectiveness. The renewal and advancement of VR technology and equipment and their integration with other cross-disciplinary disciplines have provided a new impetus for the development of virtual medicine. VR has broad prospects in medicine, which are anticipated to encourage the reform of clinical diagnosis, surgery, teaching and training, and treatment systems; however, Kamphuis et al. [24] reported that the technology of applying VR to medical education is not sufficiently mature and that it is only possible to apply AR after an empirical study proves its added value to learning. Therefore, more research is required to apply VR technology in clinical practice. The utilization of VR in the realm of medicine affords young doctors the chance to engage in simulated practice, devise treatment or palliation protocols for numerous ailments, and facilitate the advancement of surgical navigation technology. Nevertheless, this endeavor poses a challenge as well, as those physicians proficient in VR technology stand a higher chance of delivering superior medical care and spearheading novel explorations in VR-assisted disease treatment. This will be a complex and challenging task to achieve that requires cooperation between medical professionals and computer network engineers who develop and design intelligent medical systems and it would be beneficial for future generations and for innovative solutions.

### 4.2. Research Hotspots

#### 4.2.1. Clusters of Categories

It can be clearly interpreted from Figure 5 that the application research of VR in the medical field involves the integration of multiple disciplines, which combines engineering, electronic science, materials science, information science and engineering, and other emerging technologies, aiming to further explore the application potential of VR technology in fields such as surgery, neuroscience, and healthcare science.

#### 4.2.2. Burst Keywords

The research of VR applications in the medical field can be roughly divided into three periods by analyzing the burst keywords in Figure 6.

The first period was from 2012 to 2014. In this period, researchers mainly focused on VR for surgical simulation training.

VR simulation shows great potential and auspiciousness as an alternative to training surgical residents outside the operating room [27]. The use of VR simulators in medical training has increased over the past decade because they enhance trainees’ technical skills in a risk-free environment. Numerous models have been developed to provide supplementary training sessions for trainee surgeons in fields such as laparoscopic techniques [28], temporal bone surgery [29], and neurosurgery [30]. Individuals with different levels of experience can be differentiated using simulator-generated objective metrics. Experienced surgeons have shorter operation times and make fewer mistakes than novice surgeons. This suggests the potential of VR simulation as a training tool and its role in formative and terminal assessments [31].

The second period was from 2015 to 2018 when the main research focus was augmented reality surgical navigation systems.

A surgical navigation system (MIS) plays an increasingly important role in the field of modern medicine, especially for the support of minimally invasive surgery (MIS), which provides surgeons with medical images and the relative pose relationships between surgical instruments and lesions, making the surgical process more accurate and efficient [32]. With the rapid development of augmented reality technology, the surgical dynamic navigation system based on this technology is constantly improving, and so in the fields of maxillofacial surgery [33], spinal surgery [34], and neurosurgery [35], the demand for real-time accurate navigation and positioning is increasingly prominent, and the related applications are becoming more and more extensive. Aiming at this trend, Skulason et al. [36] were actively involved in the research and development of augmented reality spine navigation technology. Through the combination of intraoperative 3D imaging and instrument navigation, they successfully improved the accuracy of pedicle screw placement. One of the advantages of this technique is that it does not depend on X-ray fluoroscopy, which lays the foundation for image-guided minimally invasive treatment of the thoracic spine and is expected to further optimize the efficacy and safety of thoracic spine surgery in future clinical practice.

The third period is from 2019 to the present. The application of artificial intelligence in the medical field has increasingly gained attention [37,38], and it has begun to combine VR and deep learning to analyze complex clinical tasks. From the perspective of the time evolution path, VR research in the medical field has gradually changed from simulation training to clinical application.

Initially, AI was introduced for the processing of complex data, such as electroencephalogram (EEG) readings from patients with Alzheimer’s disease and children with attention deficit hyperactivity disorder (ADHD) in a VR environment to assess their levels of attention and emotional responses [39,40]. Subsequently, it was discovered that AI could rapidly process large volumes of images, leading to the development of an intraoperative navigation system based on deep learning (DL). This included an intelligent navigation technology that integrated AR, customized tactile surgical research tools, and lung biopsy using deep neural networks (DNNs) [41]. Simultaneously, these technologies have been applied to address more intricate clinical challenges, such as the automatic reconstruction of cranial defects through DL and implant modeling frameworks [25].

A new trend in research has emerged that combines visualization technologies like VR, AR, and MR with the Internet of Things (IoT), wireless sensor networks (WSNs), big data analysis (BDA), and cloud computing (CC) to create diverse system architectures [42,43]. The integration of these technologies is anticipated to yield innovative medical technologies for efficient and effective intelligent medical care systems. These advancements also serve as a reliable technical foundation for realizing interconnectedness within the metaverse and symbiosis between realities in the future. For instance, intelligent healthcare systems established in hospitals or medical centers will be able to provide immediate and accurate data to medical personnel so that they can provide precise diagnosis and treatment for patients.

However, the combination of visualization technologies and AI still faces many limitations and obstacles in the face of practical medical problems. For example, the validity of physiological data needs to be improved, the extensibility of methods needs to be evaluated, the current hardware capacity is limited, there is a lack of tactile feedback, data privacy issues are controversial, the application cost of some systems may beyond affordability, and there is still a gap between the progress made by neophytes in simulated operation training and clinical practice; the latter is somewhat risky but the trainers make greater progress [44,45,46,47,48]. This requires the collaboration of doctors, AI engineers, and other professionals to develop a more powerful multi-modal treatment system.

### 4.3. Present Limitations of VR in Medicine

It was found that the limitations of VR in medicine could be divided into the following two categories by summarizing the limitations of the top 10 cited studies listed in Table 3:

(1) The virtual simulation scene was still quite different from a real clinical work situation, and there was an inadequate sense of presence. In addition, many unexpected risks in surgical operations are not considered or are challenging to present on the simulator, making it difficult to fully convert the technology learned by residents on the simulator into true surgical skills.

(2) The existing virtual scenes have many types and wide coverage, such as simulating patients, emergency operations, and surgical skill training. However, these virtual scenarios are independent, indicating that they only focus on cultivating doctors’ abilities and ignore doctors’ ability to grasp the entire diagnosis and treatment process.

The metaverse development may become a solution to these problems and a new power source for the development of virtual medicine. What sets metaverse apart from other medical imaging simulation models is its incomparable number and diversity of data sets, vast social scale, and emphasis on user immersion, interaction, and collaboration [11,12]. This will give the virtual scene a stronger sense of immersion and bring it closer to clinical reality. Furthermore, a metaverse can harmoniously integrate patients, doctors, researchers, algorithms, devices, and data, improving doctors’ grasp of the diagnosis and treatment process. In addition, real-time data sharing, telemedicine systems, and virtual comparative scanning supported by metaverse systems may also be research frontiers of virtual medicine in the future.

### 4.4. Limitations of This Bibliometric Study

This bibliometric study has a few limitations. First, only articles published between 2012 and 2023 were analyzed. However, some ongoing studies have not been published yet, thus affecting this study’s prospects. Second, only articles written in English in the WoSCC academic database were analyzed. This implies that some studies may have inevitably been missed. Because it is not feasible to fuse and analyze data from different databases or languages simultaneously, articles from other databases or languages were excluded from this study. Third, inherent researcher bias cannot be ruled out, even if the 2143 articles analyzed in this study were all read.

## 5. Conclusions

Medical products incorporating VR have been used or are being used worldwide and are under continuous development. The number of publications on VR applications in the medical field has been steadily increasing year by year. The USA is the leading country in this area, while the University of London stands out as the most published, and most influential institution. VR is applied in many medical fields, focusing on anatomy teaching, surgery training, treatment of patients with neurological diseases, post-stroke rehabilitation, and telemedicine systems. It not only improves the teaching quality at the educational level but also improves the efficiency of operation and reduces the medical risk in practice. However, there are also limitations to consider. For instance, VR technology alone can only provide stereoscopic effects, offering limited assistance in addressing complex clinical issues. With the development of artificial intelligence technology, researchers combine AR and VR to make it possible to process many complex tasks quickly, assist in diagnosis, and provide diagnosis and treatment plans, indicating the robust development potential in the medical field. In the information age, there is a growing demand for the diversity, immediacy, and quality of healthcare services. Therefore, the trend of future research is to establish intelligent healthcare systems in hospitals or medical centers by integrating visualization technologies such as VR, AR, and MR with IoT, WSNs, BDA, and CC to provide medical personnel with instant and accurate data, enabling them to provide precise diagnoses and treatments for patients.

## Figures and Tables

**Figure 1 healthcare-12-01266-f001:**
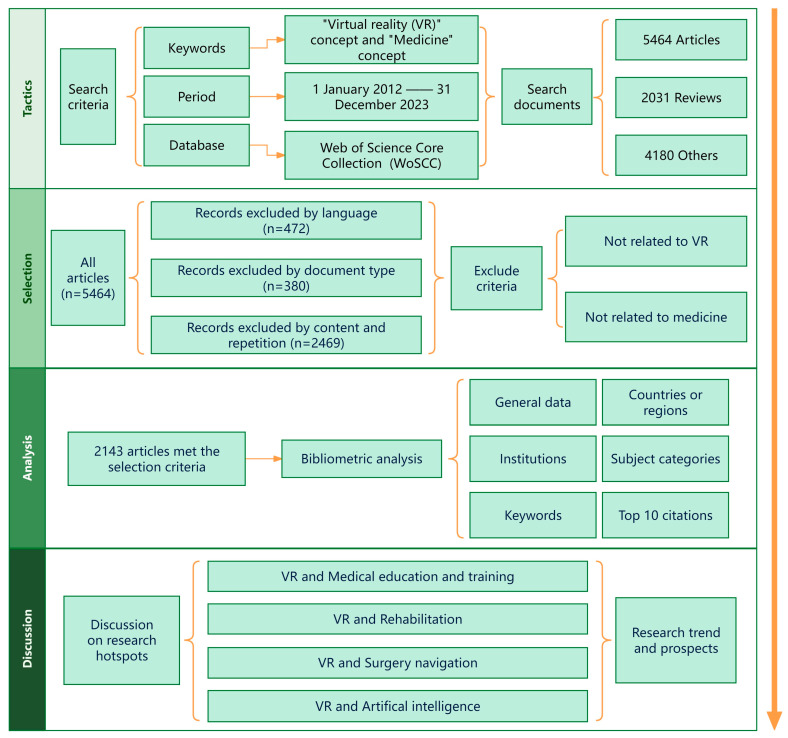
A frame flow diagram showing the specific selection criteria and bibliometric analysis steps for the study of VR in medicine between 2012 and 2023.

**Figure 2 healthcare-12-01266-f002:**
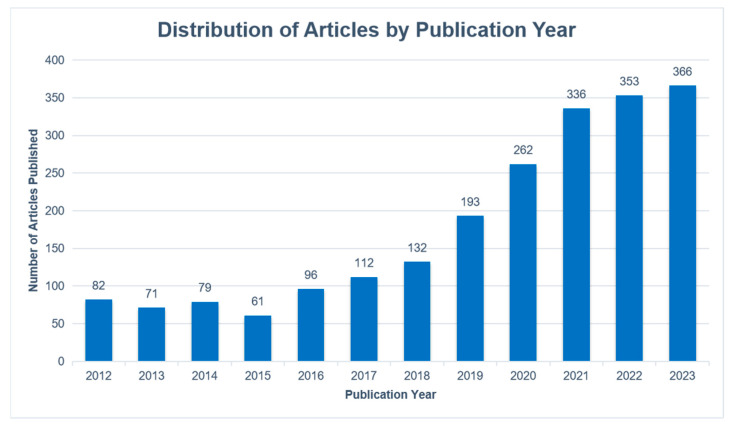
The annual number of publications on VR in medicine between 2012 and 2023.

**Figure 3 healthcare-12-01266-f003:**
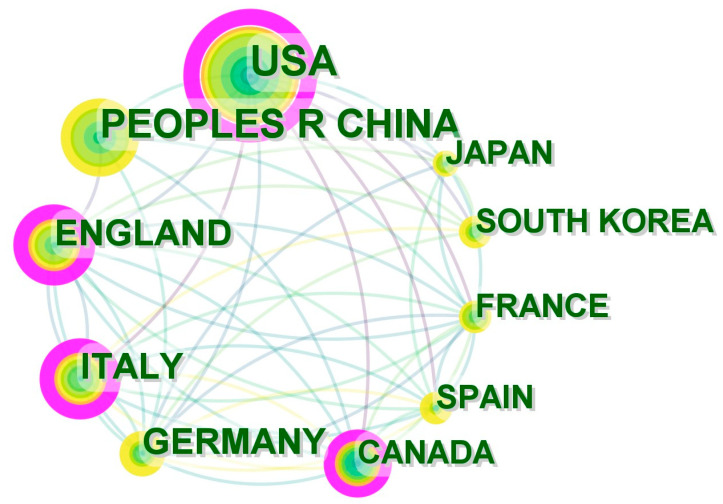
Collaboration of countries or regions that contributed to publications on VR in medicine between 2012 and 2023. Section 3.1.

**Figure 4 healthcare-12-01266-f004:**
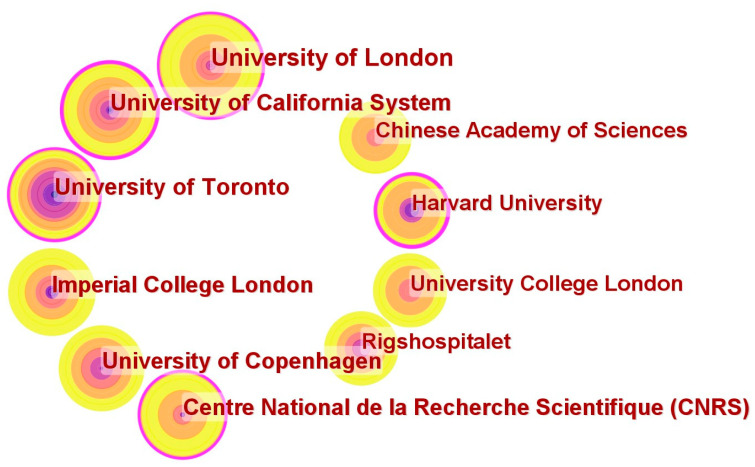
Cooperation of institutions that contributed to publications on VR in medicine between 2012 and 2023.

**Figure 5 healthcare-12-01266-f005:**
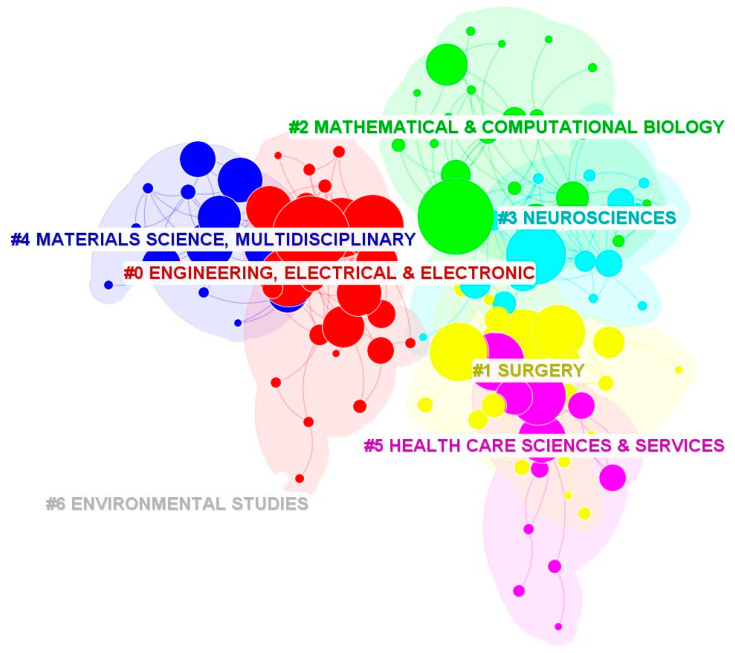
Category-based clusters of publications on VR in medicine between 2012 and 2023. The symbol means Cluster.

**Figure 6 healthcare-12-01266-f006:**
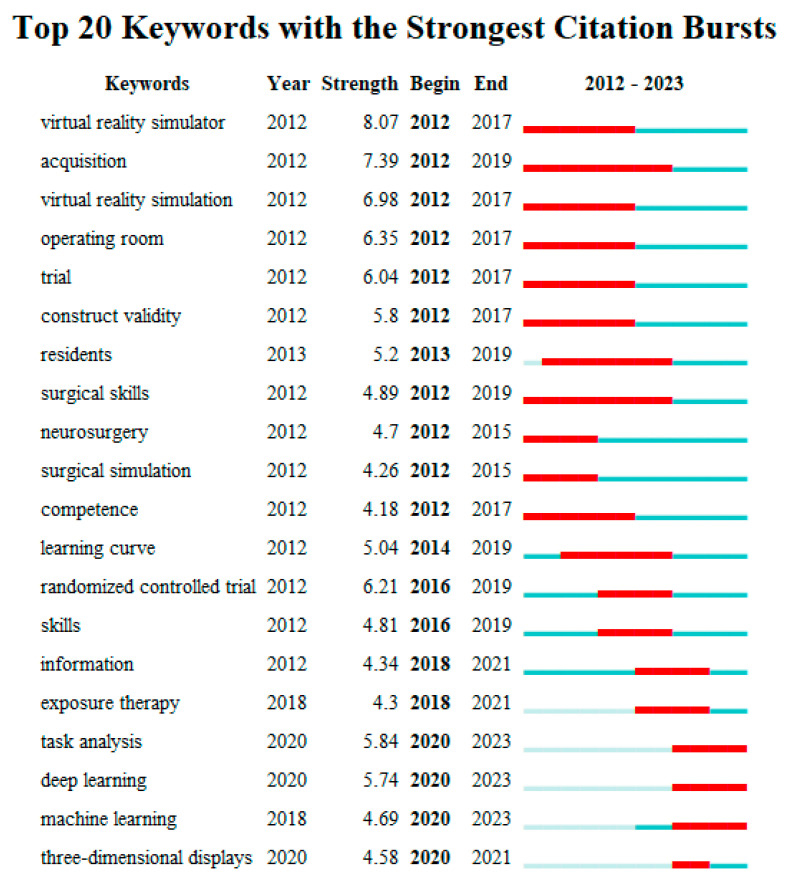
Keywords with the strongest citation bursts for publications on VR in medicine between 2012 and 2023.

**Table 1 healthcare-12-01266-t001:** Top 10 countries or regions with publications on VR in medicine between 2012 and 2023.

Rank	Country or Region	Counts	Centrality	H-Index
1	USA	461	0.41	43
2	China	329	0.08	26
3	UK	190	0.39	26
4	Italy	189	0.23	29
5	Germany	166	0.04	25
6	Canada	144	0.10	28
7	Spain	116	0.07	24
8	France	111	0.08	20
9	Republic of Korea	93	0.01	15
10	Japan	92	0.03	16

**Table 2 healthcare-12-01266-t002:** Top 10 institutions with publications on VR in medicine between 2012 and 2023.

Rank	Institution	Counts	H-Index	Country or Region
1	University of London	43	13	UK
2	University of California System	36	15	USA
3	University of Toronto	35	17	Canada
4	Imperial College London	33	12	UK
5	University of Copenhagen	33	13	Denmark
6	Centre National de la Recherche Scientifique (CNRS)	31	11	France
7	Rigshospitalet	27	11	Denmark
8	University College London	26	11	UK
9	Harvard University	26	13	USA
10	Chinese Academy of Sciences	26	9	China

**Table 3 healthcare-12-01266-t003:** Top 10 cited articles on VR in medicine between 2012 and 2023.

Rank.	Title of Citing Documents	DOI	Times Cited	Subject	Interpretation of the Findings	Research Limitations
1	Using Technology to Maintain the Education of Residents During the COVID-19 Pandemic [15]	10.1016/j.jsurg.2020.03.018	502	Training and education	Several innovative solutions using technology may help bridge the educational gap for surgical residents during the COVID-19 pandemic.	Further clinical study is needed to adequately demonstrate the usefulness of these techniques in maintaining resident education.
2	The Effectiveness of Virtual and Augmented Reality in Health Sciences and Medical Anatomy [16]	10.1002/ase.1696	421	Training and education	This paper introduces both VR and AR as potent teaching tools, where student learning is as fruitful as using tablet-based applications.	1. There is no significant difference in test scores between student learning using VR and using the traditional patterns. 2. Participants experienced side effects such as blurred vision and disorientation during the process, which affected the results.
3	The virtual reality head-mounted display Oculus Rift induces motion sickness and is sexist in its effects [17]	10.1007/s00221-016-4846-7	330	Technology assessment	The results suggest that users of contemporary head-mounted display systems are at significant risk of motion sickness. Regarding motion sickness, these systems may be sexist in their effects.	This study’s finding has little significance in guiding clinical work. The guiding significance of this study for clinical work was not obvious.
4	Extending the technology acceptance model to explore the intention to use Second Life for enhancing healthcare education [18]	10.1109/TMECH.2010.2090353	203	Training and education	This study constructs a virtual ward for nursing students to practice rapid sequence intubation.	There are still many differences between the virtual simulation of intubation and practical clinical situations.
5	Learning Anatomy through Mobile Augmented Reality: Effects on Achievement and Cognitive Load [19]	10.1002/ase.1603	187	Training and education	An AR-based anatomical learning program has been shown to be efficient in improving medical students’ learning performance and reducing cognitive load.	The sample size of this study is insufficient.
6	Direct manipulation is better than passive viewing for learning anatomy in a three-dimensional virtual reality environment [20]	10.1016/j.compedu.2016.12.009	186	Training and education	Direct manipulation of an anatomical structure in a 3D VR program benefits anatomy learning, especially for participants with low spatial ability.	Participants with greater background knowledge were more capable of effectively using the system to learn, which exerts a great influence on experimental results.
7	Using virtual reality to characterize episodic memory profiles in amnestic mild cognitive impairment and Alzheimer’s disease: Influence of active and passive encoding [21]	10.1016/j.compedu.2012.05.011	184	Clinical use	A VR test is more suitable for the early diagnosis and rehabilitation of patients with suspected Alzheimer’s disease than traditional (oral) memory tools currently available.	1. The scope of the verified data is limited. 2. The clinical value of the VR test remains to be validated by clinical trials.
8	Development of a Hand-Assist Robot With Multi-Degrees-of-Freedom for Rehabilitation Therapy [22]	10.1016/j.neuropsychologia.2011.12.013	178	Clinical use	A virtual reality (VR)-enhanced new hand rehabilitation support system that enables patients to exercise alone.	There are still many unresolved issues regarding the further clinical application of this rehabilitation support system.
9	Augmented reality in medical education? [23]	10.2196/mental.7387	157	Training and education	AR technology-supported learning makes ubiquitous, collaborative, and contextual learning possible. It provides a sense of presence, immediacy, and immersion that may benefit the learning process.	Further research with abundant samples and valid measurements remains to be conducted.
10	Basic endovascular skills for trauma course: Bridging the gap between endovascular techniques and the acute care surgeon [24]	10.1097/TA.0000000000000310	147	Training and education	A VR-based simulation for endovascular skill training.	Only 13 participants attended the test. The small sample size of this study makes the result less convincing.

## Data Availability

The original contributions presented in the study are included in the article. Further inquiries can be directed to the corresponding authors.

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
