# Peer review of "Unveiling the Evolution of Virtual Reality in Medicine: A Bibliometric Analysis of Research Hotspots and Trends over the Past 12 Years"

_healthcare, 2024, doi:10.3390/healthcare12131266_

Round 1
Reviewer 1 Report
Comments and Suggestions for Authors
The Article Unveiling the Evolution of Virtual Reality in Medicine: A Bibliometric Analysis of Research Hotspots and Trends Over the 3ast 12 Years, performs a bibliometric and content analysis of VR over the last 12 years.
The article is well written, presents a coherent methodology and interesting results; contributes to the state of the art and makes a good synthesis of it.
Despite these, there are some improvements that can help to realise its full potential, which are summarised below.
Regarding the title, although the phrase 'Unveiling the Evolution of Virtual Reality in Medicine' is an appealing one, it should be noted that the article is actually a bibliometric analysis. so the title as it is generates expectations that are not fully resolved
It is important to highlight results about topics, keywords, top cited results or phases of publication on RV above country and university. It is very interesting the information about also fields in which VR is usually published (training and education, clinical uses...). For example, Figure 6 should be highlighted as it contains the results.
In the introduction, between lines 41 to 53, add the uses in PTSD, which in mental health are very important. Also in this paragraph it would be interesting to explain and briefly cite the sub-types of VR (AR, XR...).
In method. It is not clear how the selection bias was handled, as it does not mention which of the authors made the selection. No information is given on the degree of concordance between those who made the selection or if only one author did so.
Results
Figures of the results 3 y 4 can contribute more if you point data (e.g. number of publications) or group them by sub-sets by country of publications (Fig.4). In general, tables 1 and 2 and figures 3 and 4 are redundant.
In table 3 of top cited articles add the subject (training and education, general review, interface, clinical use), as it allows an interesting grouping both as a result and for discussion and summary.
In line 320 discussions briefly define and cite a reference in relation to the sense of presence for neophytes.
In discussions add a commentary about the need of reserach regarding the affectivity of VR uses as well as the necessary competencies of professionals for their use, something not found in the article
Author Response
Comments 1:
Regarding the title, although the phrase 'Unveiling the Evolution of Virtual Reality in Medicine' is an appealing one, it should be noted that the article is actually a bibliometric analysis. so the title as it is generates expectations that are not fully resolved.
Response 1:
Thank you for pointing this out. The purpose of our article is to offer an insightful analysis of the advancements and transformations in the field of VR medicine, utilizing a bibliometric approach. The title of our article has been deliberately crafted to clearly convey its bibliometric nature. We believe that this title accurately encapsulates the research objective and methodology employed in the article. Your suggestions are greatly appreciated.
Comments 2:
It is important to highlight results about topics, keywords, top cited results or phases of publication on VR above country and university. It is very interesting the information about also fields in which VR is usually published (training and education, clinical uses...). For example, Figure 6 should be highlighted as it contains the results.
Response 2:
We fully concur with the aforementioned comment. To underscore this vital point, we have made revisions to paragraph 4.2.2. In this section, we have also given prominence to and delved into the burst keywords presented in Figure 6. Utilizing the information presented in Figure 6, our article offers an in-depth analysis and comprehensive summary of the evolving landscape of VR medicine. Furthermore, we have zeroed in on the emerging trends in VR medicine, which are evident from key terminologies such as “ask analysis”, “deep learning”, and “machine learning”. You can inspect the amendments we have made, which span from line 286 to 354.
Comments 3:
In the introduction, between lines 41 to 53, add the uses in PTSD, which in mental health are very important. Also in this paragraph it would be interesting to explain and briefly cite the sub-types of VR (AR, XR...).
Response 3:
Thank you for highlighting this matter. We concur with your observation and, accordingly, have incorporated the applications in PTSD, along with an elucidation of the various subtypes of VR, such as AR and XR. This modification is reflected in the revised manuscript, specifically within the lines ranging from 41 to 63.
Comments 4:
In method. It is not clear how the selection bias was handled, as it does not mention which of the authors made the selection. No information is given on the degree of concordance between those who made the selection or if only one author did so.
Response 4:
Thank you for your comments. Concerning your inquiries pertaining to the intricacies of the research procedure, the present paper primarily delves into the utilization of bibliometrics in elucidating research trends, rather than explicitly outlining the detailed steps of bibliometric analysis. It is noteworthy that human factors significantly influence the dissemination of documentary information, resulting in numerous challenges in quantifying document-related issues. Specifically, the inherent complexity and volatility of the literature system render it infeasible to acquire comprehensive and reliable data that could facilitate the disclosure of macroscopic patterns within the literature. The resolution of these challenges necessitates the development and application of more robust mathematical tools and statistical methodologies.
Comments 5:
Figures of the results 3 y 4 can contribute more if you point data (e.g. number of publications) or group them by sub-sets by country of publications (Fig.4). In general, tables 1 and 2 and figures 3 and 4 are redundant.
Response 5:
We appreciate your observation. Figures 3 and 4 serve as data visualization representations, effectively depicting the collaborative partnerships among various countries and institutions, along with their respective influence within the VR medicine domain. This visualization approach enables us to discern patterns and characteristics, thereby extracting valuable insights from the underlying data. However, it is noteworthy that these figures may lack specificity and precision in terms of numerical detail. Therefore, we have incorporated Tables 1 and 2, which provide a comprehensive and exact breakdown of the data, offering readers a more precise data experience compared to the visualizations alone.
Comments 6:
In table 3 of top cited articles add the subject (training and education, general review, interface, clinical use), as it allows an interesting grouping both as a result and for discussion and summary.
Response 6:
Agreed. We have incorporated the subject matter pertaining to training and education, general overview, interface, and clinical application into Table 3 of the highly cited articles. The updated version of the manuscript reflecting this alteration is now accessible in Table 3.
Comments 7:
In line 320 discussions briefly define and cite a reference in relation to the sense of presence for neophytes.
Response 7:
Thank you for pointing this out. We agree with this comment. Therefore, we have briefly defined and cited a reference in relation to the sense of presence for neophytes . This modification is reflected in the revised manuscript, specifically within the lines ranging from 350 to 352.
Comments 8:
In discussions add a commentary about the need of reserach regarding the affectivity of VR uses as well as the necessary competencies of professionals for their use, something not found in the article.
Response 8:
Thank you for your valuable feedback. We have incorporated a commentary into the revised manuscript addressing the necessity of further research into the effectiveness of VR applications, as well as the essential competencies required by professionals for their utilization in discussions. This modification can be located between lines 268 and 274.
Reviewer 2 Report
Comments and Suggestions for Authors
More relevant literatures need to be added to the manuscript

Author Response
Comments 1:
Since the article is submitted to a healthcare journal, it would add value if the authors could provide a more detailed categorization of the use of VR/AR/MR or extended reality technology in healthcare practice.
Response 1:
We appreciate your valuable feedback on this matter. Having carefully considered your comment, we are in complete agreement with your observation. Consequently, we have endeavored to furnish a more meticulous categorization of the utilization of VR/AR/MR or extended reality technology in healthcare practice. This revision can be located in the manuscript, specifically between lines 50 and 55.
Comments 2:
The introduction would benefit from additional relevant literature, emphasizing the gap in the existing literature and the need for the current study. The primary outcome of this study should be explicitly stated.
Response 2:
Thank you for your comments. There remains a noteworthy discrepancy between the present endeavors and anticipated future trends, thereby posing novel challenges and offering fresh avenues for the advancement and utilization of VR technology within the medical domain. To underscore this disparity in the existing corpus of literature and highlight the necessity of the present study, we have incorporated pertinent references. This alteration can be located in the revised manuscript, specifically spanning from line 58 to line 63.
Comments 3:
The process used by the authors to rank the articles needs to be explained.
Response 3:
Thank you for pointing this out. The specific process has been written in detail in figure1.
Comments 4:
The graphical representations in figures 3 and 4 are new but lack adequate explanation. The authors should clarify what each hue/saturation of each color on the rings specifies.
Response 4:
Thank you for your comments. We have added the clarification of what each hue/saturation of each color on the rings specifies in figures 3 and 4. In the revised manuscript this change can be found between line 112 to 119.
Comments 5:
In Table 2, the reason for underlining a few universities is not clear.
Response 5:
Thank you for pointing this out. Underlining a few universities is an accident in the process of text conversion, and there is no intention to emphasize any institution here.
Comments 6:
Additionally, it would be helpful to have additional quantitative figures from the clusters in Figure 5. The authors should explain how they ranked the articles in Table 3, both in the methods and results sections.
Response 6:
Figure 5 is a data visualization image. Compared with the specific numerical value, it vividly reflects the interaction between various technologies involved in VR in the medical field, thus discovering the laws and characteristics. We also explain the sorting rules of the articles in Table 3. In the revised manuscript this change can be found between line 232 to 233.
Comments 7:
Comparing the citations of Virtual Reality, Augmented Reality, and Extended Reality with respect to healthcare applications would be useful.The use of VR/AR/XR should be discussed with respect to different categories of healthcare.
Response 7:
We have provided a detailed explanation and concise citation of the various subtypes of VR (including AR, XR, and others) pertaining to their healthcare applications. This modification can be located in the revised manuscript, specifically between lines 50 and 55.
Comments 8:
The conclusion should include the main findings from the study, which need to be strong and relevant to the motivation of the study. The authors should consider moving the last line of the conclusion to the discussion section. Suggestion: Despite its complexity, the authors could mention that an integrated approach from medical professionals and engineers to healthcare would be beneficial for future generations and for innovative solutions.
Response 8:
We concur with the suggestion and have accordingly relocated the final line of the conclusion section to the discussion section. Additionally, we have emphasized the significance of a collaborative effort between medical professionals and engineers in the realm of healthcare, as it holds immense potential for fostering advancements and innovative solutions that will greatly benefit future generations. This modification can be traced in the revised manuscript, specifically from line 274 to 277.
Reviewer 3 Report
Comments and Suggestions for Authors
In an era of technological evolution, it is essential to understand this evolution and impact on medical practice, congratulations on the topic. The analysis carried out only aims to clarify aspects that do not seem adequately described but that can certainly be easily integrated.
The methodology used in this review is not clear in the abstract, is it just a literature review?
In the methodology section, doubts remain. The objective of the study is not clear and the choice of only the web of science with database, the search phrase/words do not seem to be enlightening for the results identified.
The results are interesting in the way they are presented and in their content, but the process of arriving at them does not seem clear to me.
Author Response
In an era of technological evolution, it is essential to understand this evolution and impact on medical practice, congratulations on the topic. The analysis carried out only aims to clarify aspects that do not seem adequately described but that can certainly be easily integrated.
The methodology used in this review is not clear in the abstract, is it just a literature review?
In the methodology section, doubts remain. The objective of the study is not clear and the choice of only the web of science with database, the search phrase/words do not seem to be enlightening for the results identified.
The results are interesting in the way they are presented and in their content, but the process of arriving at them does not seem clear to me.
Dear reviewer:
We appreciate your valuable comments.
Our paper is a bibliometric study that differs substantially from a literature review. Bibliometrics represents a research approach that employs quantitative analysis in order to assess the scientific literature, with a particular emphasis on the evaluation of publication and citation data. The primary objective of bibliometrics is to acquire a comprehensive understanding of the underlying structure, evolution, and dynamics of scientific disciplines. Bibliometric analysis primarily focuses on citation analysis, publication analysis, co-authorship analysis, journal analysis, and research trend analysis. Citation analysis, specifically, involves examining the citation patterns of various publications in order to assess their influence within the scientific community.
The decision to select the literature from the Web of Science database as the analytical dataset was based on its established authority, the consistently high quality of its contents, its notable representativeness, and its ability to effectively mirror prevailing research trends.
Regarding search phrase/words, since bibliometrics articles identify emerging themes and trends in research mainly by analyzing the frequency and patterns of keywords, subject categories, or author affiliations in the published literature. So we combined the key topics in VR to make up our search mode.
Concerning your inquiries pertaining to the intricacies of the research procedure, the present paper primarily delves into the utilization of bibliometrics in elucidating research trends, rather than explicitly outlining the detailed steps of bibliometric analysis. It is noteworthy that human factors significantly influence the dissemination of documentary information, resulting in numerous challenges in quantifying document-related issues. Specifically, the inherent complexity and volatility of the literature system render it infeasible to acquire comprehensive and reliable data that could facilitate the disclosure of macroscopic patterns within the literature. The resolution of these challenges necessitates the development and application of more robust mathematical tools and statistical methodologies.